# Enhancing the Thermal Conductivity of Amorphous Carbon with Nanowires and Nanotubes

**DOI:** 10.3390/nano12162835

**Published:** 2022-08-18

**Authors:** Geraudys Mora-Barzaga, Felipe J. Valencia, Matías I. Carrasco, Rafael I. González, Martín G. Parlanti, Enrique N. Miranda, Eduardo M. Bringa

**Affiliations:** 1Consejo Nacional de Investigaciones Científicas y Técnicas (CONICET), Mendoza 5500, Argentina; 2Facultad de Ingeniería, Universidad de Mendoza, Mendoza 5500, Argentina; 3Departamento de Computación e Industrias, Facultad de Ciencias de la Ingeniería, Universidad Católica del Maule, Talca 3480112, Chile; 4Centro para el Desarrollo de la Nanociencia y la Nanotecnología (CEDENNA), Avda. Ecuador 3493, Santiago 9170124, Chile; 5Escuela de Ingeniería Industrial, Facultad de Ciencias, Universidad Mayor, Santiago 8580745, Chile; 6Center for Applied Nanotechnology, Universidad Mayor, Santiago 8580745, Chile; 7Facultad de Ciencias Exactas y Naturales, Universidad Nacional de Cuyo, Mendoza 5500, Argentina

**Keywords:** amorphous carbon, thermal conductivity, molecular dynamics, nanowires, nanotubes

## Abstract

The thermal conductivity of nanostructures can be obtained using atomistic classical Molecular Dynamics (MD) simulations, particularly for semiconductors where there is no significant contribution from electrons to thermal conduction. In this work, we obtain and analyze the thermal conductivity of amorphous carbon (aC) nanowires (NW) with a 2 nm radius and aC nanotubes (NT) with 0.5, 1 and 1.3 nm internal radii and a 2 nm external radius. The behavior of thermal conductivity with internal radii, temperature and density (related to different levels of $sp^{3}$ hybridization), is compared with experimental results from the literature. Reasonable agreement is found between our modeling results and the experiments for aC films. In addition, in our simulations, the bulk conductivity is lower than the NW conductivity, which in turn is lower than the NT conductivity. NTs thermal conductivity can be tailored as a function of the wall thickness, which surprisingly increases when the wall thickness decreases. While the vibrational density of states (VDOS) is similar for bulk, NW and NT, the elastic modulus is sensitive to the geometrical parameters, which can explain the enhanced thermal conductivity observed for the simulated nanostructures.

## 1. Introduction

Over the last decades, many researchers have studied the thermal transport properties in amorphous materials [1,2,3,4,5,6,7] due to their use as thermal devices in a wide range of emerging applications, such as energy-efficient construction, thermal energy storage, energy dissipation and aerospace applications [8,9,10,11,12]. Amorphous materials, which are typically promising structures for flexible electronics, MEM resistive devices or protective coatings [13,14], can be compromised in their application due to their intrinsic low thermal conductivity, which can promote, for instance, thermal cracking.

Several approaches have been used to improve the thermal conductivity of amorphous materials, as is the case of polymers, amorphous silica and diamond-like carbon, among others. In search of a better thermal insulator, great efforts have been devoted to designing novel materials [15], including porous structures. The presence of pores decreases the overall conductivity, such as in porous aerogels based on carbonaceous materials (carbon nanofibers, graphene, etc.) [10,16,17] and polymers (cellulose and Styrofoam) [18,19].

Recently, it was shown that an oxide with nanoscale porosity can display ultra-low thermal conductivity [20]. From a structural point of view, porous structures play a crucial role in thermal insulation and mechanical properties [1,21]. Thermal conductivity for systems with reduced dimensionality can also display unusual behavior [22,23]. For instance, anomalous high thermal conductivity was reported in amorphous Si thin films [24].

Enhanced thermal conductivity has been reported in carbon-based composites with nanotubes [25,26]. Of course, nanomaterials improving thermal conductivity are typically affected by several factors [8], and finding the optimal combination of size, shape and/or pore distributions appears to be the key in balancing thermal properties. Generally, a pore distribution introduces barriers for thermal transport but also compromises the mechanical strength of the material. This conflict between thermal insulation and mechanical resistance caused by pore structures is primarily responsible for the unwanted mechanical properties of most aerogels [10].

Nature is an expert in testing and designing structures with optimal properties. An interesting example is the tubular hair of polar bears, an excellent thermal insulator that helps polar bears survive cold climates [10,11]. Biomimetic design can lead to improved synthetic insulators. Unlike the hairs of humans or other mammals, the hairs of polar bears are hollow. These cavities have remarkable heat retention capacity and elasticity. To emulate this structure, Zhan and collaborators [10] manufactured hollow carbon tubes, each equivalent to a single strand of hair, and compacted them in a spaghetti-type aerogel block.

Compared to other aerogels and insulation components, the researchers found that the hollow tube design inspired by the polar bear was lighter and more resistant to heat flow. As an added benefit, the new material was remarkably elastic. Contrary to this principle, carbon nanotubes exhibit extremely high thermal conductivity along the tube axis [27]; however, their transport properties can be reduced if the thermal gradient is perpendicular to their axis [28]. All in all, effects, such as the nanotube size, wall structure and thickness and thermal gradient orientation are factors that strongly influence the thermal behavior of tubular nanomaterials.

The conductivity of amorphous carbon samples is orders of magnitude lower than the conductivity of single crystal diamond or the in-plane conductivity of graphite. However, the cross-plane conductivity of graphite or the conductivity of ultra-nanocrystalline diamond (UNCD) is nearly as low as the conductivity of some amorphous C samples [29]. Experimental results show that thermal conductivity in aC depends mainly on the amount and the structural arrangement of the $sp^{3}$ phase. If the $sp^{3}$ phase is amorphous, the thermal conductivity *k* is linear with the $sp^{3}$ content, the density and the elastic modulus [29,30,31,32,33].

Conductivity experiments in aC give conductivities below 5 Wm${}^{-1}$K${}^{-1}$; however, results for amorphous diamond films give values between 5 and 10 Wm${}^{-1}$K${}^{-1}$ [34]. Most measurements of aC thermal conductivity are conducted on films. However, we are not aware of experimental results for the thermal conductivity of amorphous carbon (aC) nanowires or nanotubes. There are atomic-scale simulations for aSi bulk [35], NW [3,36,37] and NT [38,39]; however, the aC thermal conductivity has been simulated only for bulk.

There are simulations for a single high $sp^{3}$ content (tetrahedral aC-taC-), with a density of 3.0 g/cm${}^{3}$ [40,41]. Zhang et al. [42] simulated an aC film deposited on a diamond substrate, with several $sp^{3}$ contents below 25%, using the Tersoff potential [43]. Suarez-Martinez and Marks [44] used the Environment-Dependent Interaction Potential (EDIP) [45] and generated amorphous samples similar to the ones used here, with densities in the range 1.4–3.0 g/cm${}^{3}$. They employed the equilibrium Müller–Plathe method [46] to obtain room temperature thermal conductivities showing a linear dependence with density in good agreement with experiments on aC films [47].

Recently, amorphous bulk samples reaching 3.31 g/cm${}^{3}$ (76% $sp^{3}$) were simulated using an extended Tersoff potential and lattice dynamics. For densities above 2.8 g/cm${}^{3}$, propagons played a dominant role due to the increased atomic coordination [48].

Motivated by the experimental work of Zhan et al. [10], we explore the thermal conductivity of a single, nanoscale, amorphous carbon hollow tube, which could help understand future experimental results. We present a molecular dynamics study of the thermal conductivity of amorphous carbon nanowires (NW) and nanotubes (NT). We obtain the conductivity versus temperature and analyze the roles of hybridization, the density, the internal radii and the elastic modulus of the nanostructures.

## 2. Methods

### 2.1. Non-Equilibrium MD (NEMD) Setup

Simulations were performed with the software LAMMPS [49] using the EDIP interactions [45], which can simultaneously describe the aC density, hybridization and microstructure well compared with experiments [50,51]. Visualization, analysis and data post-processing were performed using the Open Visualization Tool (OVITO) software [52].

Amorphous carbon (aC) bulk samples with the desired $sp^{3}$ hybridization are obtained following the heat treatment proposed by Tomas et al. [50], quenching molten carbon from 5000 to 300 K using a linear temperature ramp of 0.5 ps. The resulting amorphous carbon bulk density was adjusted by setting the initial number of atoms in the simulation cell, before melting and subsequent quenching. The bonds of each carbon atom, and therefore the $sp^{3}$ contents of the samples, are defined using a coordination criterion.

Coordination is calculated based on a radial cut-off of 0.2 nm, which only includes the first neighbors according to the pair correlation function g(r). A carbon atom is $sp^{2}$ if there are three carbon atoms within a 0.2 nm cutoff radius and $sp^{3}$ if there are four carbon atoms within the same cutoff radius. This is a standard approach used in classical molecular dynamics where there are no electronic orbitals to define hybridization [45].

The NW and NT with different $sp^{3}$ concentrations are generated by cutting the aC bulk samples with 20%, 30% and 55% of $sp^{3}$. Due to the artificial cut, the resulting surface stresses are relaxed using the conjugate gradient algorithm in LAMMPS. This is followed by an annealing at 1200 K during 5 ps and finally cooling the sample down to 300 K. Bulk samples of different densities have slightly different dimensions along z (main axis), as a result of sample relaxation. Therefore, the NW and NT lengths also vary slightly with $sp^{3}$ content. Only NT with 2 nm external radius are studied here, varying their internal radius up to 1.3 nm and a minimum thickness of 0.7 nm. Smaller thicknesses are not considered because they lead to the formation of pores in the NT wall and a considerable change in the $sp^{3}$ concentration.

To set up the system for the thermal conductivity measurement, first, all nanowire and nanotube samples are cut from bulk samples at 300 K. The whole sample is then taken to the desired temperature with a linear ramp lasting 0.25 ns, then equilibrated at that temperature during an additional time of 0.25 ns. Then, the two regions at the extremes of the NW/NT/bulk, 0.65 nm thick, are selected, and the atoms there are fixed during the rest of the simulation. After this, adjacent to these fixed regions, two regions with a width of 1 nm are selected, and their temperatures are set to 0.8T and 1.2T (Figure 1) to force a thermal gradient along the NW/NT/bulk.

This temperature gradient is applied for 2.5 ns until it is nearly constant throughout the system (Figure 2). Thermodynamic properties of the system are calculated every 0.5 ps during the following 0.5 ns. The averages of these measurements are used to calculate the thermal conductivity as explained in Section 2.2. The total simulation time is 3.5 ns for each case, using a time step of 0.5 fs. For each radius, samples were studied at 50, 100, 200, 300, 400, 500 and 600 K.

The structural integrity was verified during the synthesis and equilibration of the samples as well as during the measurement of their thermal conductivity. We monitored coordination and also the possible formation of pores in the nanotubes, without finding any significant changes. There was no buckling of the samples for the chosen aspect ratios and boundary conditions. During the simulations, we carefully checked that the thermal gradient did not create any significant shear or volumetric strains in the sample. The average surface roughness of our samples was characterized by its RMS (Root Mean Square) roughness
$$\delta_{\mathrm{RMS}}=\sqrt{\frac{1}{N}\sum^{N}{\left(r_{i}-r_{m}\right)}^{2}}$$
where $r_{i}$ is the radial distance of the atom to the z axis, and $r_{m}$ is the average $r_{i}$, summing over all atoms within 0.2 nm of the surface. This was found to be 0.06 nm or below and is not expected to significantly affect the conductivity [53,54].

### 2.2. Thermal Conductivity Calculation

In crystalline solids, it is common to find a decrease in the lattice thermal conductivity as the temperature increases due to growing disorder affecting the coherence of phonons and decreasing their mean free path. However, for an amorphous material, the lack of symmetry in the material means that there are no phonons but vibrational modes transporting energy. The average mean free path of these modes is of the order of the distance between first neighbors, which varies relatively little with the increase in temperature if it is well below the melting temperature.

Therefore, it is usual to find a weak dependency between conductivity and temperature in amorphous materials [55]. There are several approaches to calculate thermal conductivity of nanosystems [4]. Relatively small systems, such as bulk amorphous systems, are often analyzed with Green–Kubo (GK) modal analysis [56]. This analysis allows for a detailed study of conduction modes and, for amorphous solids, includes the possibility to separate the contributions from propagons and diffusons [40,48].

These methods become too computationally expensive for large samples and provide results comparable to non-equilibrium methods. For instance, this comparison was done recently by Giri et al. for bulk aC, finding slightly lower conductivity for their GK calculations [48]. Since our samples are extremely inhomogeneous and far from a bulk sample, for the relatively large nanostructures studied here we use a non-equilibrium approach, that has been used to study nanoscale defects impact on thermal properties [57] and obtain the thermal conductivity, $k_{MD}$, by applying Fourier’s Law:(1)$$k_{MD}=J\Delta z/\Delta T$$
where $J=Q/(A_{t}t)$ and ΔT/Δz are the heat flux and the temperature gradient across the system (Figure 2), ($A_{t}$, *Q* and *t* are the thermostated area, the heat supplied by the thermostat and the time, while ΔT and Δz are the difference in temperature of the thermostats and the distance between them). The Langevin thermostat enables the calculation of the cumulative energy added/subtracted to the atoms as they are thermostated. Then, *Q* can be calculated as the average between the energy added and subtracted from the hot and cold thermostat, respectively. Figure 3 shows how a typical heat flux stabilizes after ≈30 ps.

The area used to calculate the heat flux is not easy to determine, since the NT and NW relax after their initial cut. The geometric area of the NT/NW section constitutes a reasonable approximation, as used by Wingert et al. [38]. Still, here we used in our calculations the area obtained as follows. The OVITO “SurfaceMesh” tool [52] was used to calculate the solid volume of the sample, with a test sphere parameter of 0.45 nm, which is sufficient to avoid gaps in the resulting mesh structure.

Using smaller radii (down to 0.3 nm) lead to a few artificial pores; however, changes in the surface-to-volume ratio were smaller than 5%. Then, given that buckling and bending do not occur, the effective cross-sectional area is obtained by dividing the solid volume by the length of the box along the wire axis. *T* and *Q* errors from the MD time series are about 5–10%.

The most significant error appears in the cross-sectional area calculation, being greater in the NT than the NW due to the existence of two surfaces instead of one and reaching approximately 7–15%. For some cases, we ran a few different initial velocities to set up the conductivity measurement and did not find any difference with the already calculated values within error bars.

### 2.3. The Einstein Thermal Conductivity Limit

The equation for the thermal conductivity of an ideal gas is [58]:(2)$$k=\frac{1}{3}C_{v}lv$$
where *l* is the mean free path (MFP) and *v* is the speed of sound. Although this is a simple approach, it has been used to explain experimental thermal conductivity in Si NW and NT [38]. Assuming the mean free path equal to half the wavelength of the single-frequency phonons in Equation (Equation 2), Einstein calculated the lowest possible thermal conductivity of a solid. This model, known as the Einstein limit, has been used to successfully predict the thermal conductivity of amorphous solids as [59]:(3)$$k_{Eins}=\frac{k_{B}^{2}}{\hbar }\frac{n^{1/3}}{\pi }\Theta_{E}\frac{x^{2}e^{x}}{{\left(e^{x}-1\right)}^{2}}$$
where *n* is the number density of atoms, $\Theta_{E}=\sqrt[{3}]{\frac{\pi }{6}}\Theta_{D}$ are the Einstein and Debye temperatures, which according to Wei et al. [60] is $\Theta_{D}=332.7\;K$ for aC and $x=\Theta_{E}/T$. In this work, the Einstein thermal conductivity limit is calculated and compared with our results.

### 2.4. Quantum Corrections

It is important to notice that classical molecular dynamics calculations obtain a classical specific heat ${\left(C_{v}\right)}_{C}$, without quantization of vibrational modes, nor bosonic statistics of vibrations which cause a decrease of the specific heat at low temperatures [61]. The classical simulated thermal conductivity is nearly constant with temperature. There are several proposed methods to include quantum corrections to these values. For instance, one can multiply by the ratio between the quantum and classical specific heat, ${\left(C_{v}\right)}_{Q}/{\left(C_{v}\right)}_{C}$ [62,63].

This approach has been used for disordered materials [64], although it has limitations [65]. There are alternative corrections [66,67,68], some involving separate corrections for each vibrational mode [40,69]. We apply a simple correction using the Einstein specific heat [70] and the Dulong–Petit specific heat. Within the simplistic Einstein model, there is only a single relevant frequency, and the correction can be applied without summing over all the different vibrational modes.

This leads to qualitative values for low temperatures; however, they are indicative that a conductivity decrease is expected as temperature decreases. They also indicate that quantum corrections would be relatively low at room temperature, being around 5% for this simple approximation. The thermal conductivity values reported in this work include this correction.

### 2.5. Vibrational Density of States (VDOS)

Previous work has shown the importance of the VDOS in analyzing the thermal properties in crystalline and amorphous materials [5,7,17,56,71,72,73,74,75,76,77]. This can be calculated by a Fourier transformation of the atomic velocity autocorrelation function, $Z_{\alpha }\left(t\right)$, for a sample equilibrated at the desired temperature:(4)$$G\left(\omega \right)=\int_{0}^{\infty }Z_{\alpha }\left(t\right)\cos\left(\omega t\right)dt$$
where G(ω) is the vibrational density of state at frequency ω, and $Z_{\alpha }\left(t\right)$ is given by:(5)$$Z_{\alpha }\left(t\right)=\frac{1}{N}\sum_{i}^{N}\frac{v_{i}\left(t\right)\;\cdot \;v_{i}\left(0\right)}{v_{i}\left(0\right)\;\cdot \;v_{i}\left(0\right)}$$
where $v_{i}\left(t\right)$ is the velocity of atom *i* at time *t*, and the sum is performed over all atoms in the system.

## 3. Results and Discussion

### 3.1. Structural Properties

We aim to model a relatively wide hybridization range in our simulations. EDIP is expected to deliver a reasonable agreement with experiments for bulk systems [50] regarding density versus hybridization. This was also recently discussed for large bulk aC samples [51] using EDIP, including $sp^{1}$, $sp^{2}$ and $sp^{3}$ hybridization fractions and is indeed the case for our bulk samples, as shown in Figure 4, with excellent agreement between our bulk samples and the linear relationship found by Ferrari et al. [32] for aC samples.

It can also be seen that, for each particular geometry, the relationship between the density and the percentage of $sp^{3}$ hybridization remains linear but, as expected, the generation of surfaces decreases the amount of $sp^{3}$ content. Balandin et al. [31] experimentally found that density is related to the elastic modulus of aC, and using the relationship previously obtained by Ferrari et al. [32], another linear relationship between elastic modulus and $sp^{3}$ content can be derived.

Density and elastic modulus determine sound speed, which is directly related to thermal conductivity, as in Equation (Equation 2). Some carbon-based materials might have a complex relationship between the thermal conductivity and elastic modulus [78].

The elastic modulus of a nanowire can have a complex dependence with the NW diameter [79]. Models including surface elasticity can reproduce experimental results showing modulus increases with NW diameter for crystalline Si NW or decreases with NW diameter for ZnO NW [80]. Most of the studies of NW elastic properties have been performed for crystalline NW; however, there are studies for metallic glass NW [81] and silica NW [81,82] showing strong surface effects. The NW surface can play a significant role. Surface atoms have fewer neighbors, which could lead to softening; however, bond-order potentials, such as the one used here, can lead to much stronger bonds for surface atoms.

Models considering surface elasticity often consider different elastic properties for core and surface, given some kind of “core-shell” model for NW [83,84]. Often, it is assumed that the core has the same elastic properties as the bulk. However, the core might be compressed to compensate for the surface tensile stress, and this can lead to non-linear elastic effects and an effective modulus, which varies with the NW size [79].

The interatomic potential used here provides a good description of both *E* and ρ for aC, as shown in a recent study combining experiments and simulations of nanoindentation [51]. For the NW of crystalline metals, the elastic modulus is usually independent of size, although the onset of plasticity and yield stress do strongly depend on size [85]. On the other hand, for crystalline Si NW, elastic softening of the NW was found with respect to bulk [38].

The elastic modulus was calculated from our MD simulations using the uniaxial deformation of the samples, at a strain rate of ≈10^8^/s, from the slope of the stress–strain curves at low strain (≈1%) to avoid plasticity. Figure 5 shows that our bulk and NW results are in good agreement with the relationship found by Ferrari [86], $E=478.5{\left(0.4+sp^{3}\right)}^{3/2}$. As also shown in Figure 5, the elastic modulus for NW at the nanoscale is somewhat higher than for the bulk sample.

For this reason, the Young’s modulus value could contribute to an increase in their thermal conductivity, with respect to the bulk conductivity. The aC nanotubes (1 nm inner radius) also show an increase in elastic modulus compared to bulk and to NW. The changes in the resulting velocities are of the order of 7–15% and, therefore, can account for some of the increase in thermal conductivity regarding bulk values.

We note that the Tersoff potential used by Zhang et al. [42] gives higher values of elastic modulus, which also results in higher conductivity values. For instance, the Tersoff elastic modulus is 788 GPa for 30% $sp^{3}$ [87] and, considering the appropriate density, this would give an approximate increase of conductivity of ∼1.7 with respect to EDIP, consistent with the high value of 4 Wm${}^{-1}$K${}^{-1}$ at 25% $sp^{3}$ from Zhang et al. [42].

Something similar to this has been observed in nanofoams, which are often considered as a NW array, is that the surface-to-volume ratio (S/V) controls this effect, since the core pressure can be approximated by the capillary pressure [88]. In this work, we are aiming to understand the changes in thermal conductivity, not the details in the mechanical properties of NW and NT. We consider NT geometries that go beyond the usual NW core-shell models.

However, we can take advantage of the models using (S/V), which increases going from NW to NT and increasing the inner radius of the NT. This would give a change in the elastic modulus proportional to (S/V). Figure 6 shows that this is indeed the case and that the modulus increase can be associated with higher surface contribution. Studies on Au nanofoams [89] cover a range of S/V values reaching almost 2/nm, larger than the largest value achieved here for aC NT, 0.5/nm.

### 3.2. Thermal Conductivity

Figure 7 shows the relationship between the thermal conductivity and the sound velocity calculated as $v=\sqrt{E/\rho }$, where ρ is the density of the system. The results from aC thin-film experiments by Balandin et al. [31] are also shown, where k=−2.82+1.77ρ and $k=0.04E^{2/3}-0.4$ were used. Our results are greater than the predicted by Balandin et al. [31], however, we note the linear dependency between *k* and *v*, consistent with Equation (Equation 2).

In order to fit our simulation results, we borrow the Einstein specific heat and the Mean Free Path (MFP) from the Einstein model, take *v* as $\sqrt{E/\rho }$ and replace them in Equation (Equation 2), thereby, obtaining a modifying version of Equation (Equation 3):(6)$$k_{Efit}=k_{B}n^{2/3}\sqrt{\frac{E}{\rho }}\frac{x^{2}e^{x}}{{\left(e^{x}-1\right)}^{2}}$$

Figure 8 shows the results for the thermal conductivity of NW with a radius of 2 nm and different $sp^{3}$ content, including a comparison with the Einstein conductivity model, Equations (Equation 3) and (Equation 6). As Figure 8 show, quantum corrections would be significant only below the Debye temperature [90,91]. Conductivity is roughly constant with temperature and increases with $sp^{3}$ content, as expected. In addition to aC conductivity having a relatively small temperature dependence, it has been shown that surface scattering in NW and NT leads to nearly constant thermal conductivity in Si NW [92]. Li et al. [93] also found a small dependency with temperature from 100 to 300 K in aSi using a unified deep neural network potential.

Figure 8 and Figure 9 show the dependence of the thermal conductivity on the temperature for NW and NT of 2 nm external radius and 1 nm internal radius in the NT case. For various hybridization levels, the results of Equation (Equation 6) were added for each case and show reasonable agreement with our results. In both Figure 8 and Figure 9 is shown that the thermal conductivity does not change with temperature above 200 K and increases with the percentage of $sp^{3}$ hybridization in agreement with Balandin et al. [31] for tetrahedral aC films and Shamsa et al. [47] for aC.

Dependence of the thermal conductivity with temperature, for NT of various internal radii, is shown in Figure 10; red dots represent the 2 nm radius NW, which corresponds to an internal radius of 0 nm. Experiments on aSi nanotubes show that the conductivity increases with the thickness of the nanotube.

However, those nanotubes are significantly larger than the ones studied here, since the smallest thickness is 5 nm, and the smallest diameter is about 40 nm [38]. In addition, NW and NT conductivities were found to be significantly smaller than bulk values, due to surface scattering and also to possible elastic softening. This was supported by equilibrium MD simulations of crystalline Si NT [38]. The conductivity of a MD simulated small Si NT was found to be smaller than that for the corresponding Si NW [38].

The thermal conductivity of a Si NW was shown to be higher than the bulk values, in early MD for small NW [36]. Malhotra and Maldovan [54] have studied the thermal conductivity of crystalline Si NW and NT, focusing on the decrease due to surface scattering modification of the phonon MFP and found that roughness of the inner and outer interfaces strongly decreases the conductivity mainly due to the effect on the phonon MFP.

However, Liu and collaborators [94], studying the effect of surface amorphization and roughness in single-crystalline Si NW, found that, although amorphous surface significantly reduces the thermal conductivity of Si NW due to strong phonon scattering at the interface as well as the non-propagating diffusion of phonons in the amorphous region, the thermal conductivity value of the amorphous-shell Si NW was not sensitive to roughness or vacancy in the amorphous surface.

Given the amorphous nature of our system, the phonon MFP is small, and the roughness of the surface is small enough to not have a significant effect on the heat transport, which could explain the discrepancies between the conductivity behavior reported for crystalline Si NW and our results. The results of Equation (Equation 6) show good agreement with our results, predicting the increment in thermal conductivity as the internal radius increases, as mentioned previously in Section 3.1, where a model based on S/V, can explain this behavior base on the enhanced elastic module.

Figure 11 shows the dependence of the thermal conductivity at 300 K with the percentage of $sp^{3}$ hybridization, for bulk, NW and NT of external radius 2 nm and internal radius 1 nm. Dotted lines show that the MD results follow a linear relationship with $sp^{3}$ content for all geometries. We also include the only MD value from Suarez-Martinez and Marks [44], for which hybridization is explicitly reported. For aC nanofilms thermal conductivity is linear with density [31] and, in turn, density is linear with $sp^{3}$ content [32]. Therefore, a linear dependence is expected for thermal conductivity vs. $sp^{3}$ content. In particular, we have that k(300 K)=−2.82+1.77ρ [31] and $\rho \left(300\;K\right)=1.92+1.37\;sp^{3}$ [32], resulting in $k\left(300\;K\right)=0.58+2.425\;sp^{3}$.

This fit is also shown as a dashed line in Figure 11. For the bulk MD values, the fit gives $k\left(300\;K\right)=0.94+5.42\;sp^{3}$, doubling the slope from film experiments. Zhang et al. [42] report $k\left(300\;K\right)=2.61+7.83\;sp^{3}$ for their MD simulations of films with less than 25% $sp^{3}$, using the Tersoff potential. We note that the fit by Balandin et al. [31] can show some inconsistency for carbon densities below 1.6 g/cm³ leading to zero or even negative values for *k*.

These fits have to be considered for carbon densities consistent with a homogeneous hydrogen-free, $sp^{3}$/$sp^{2}$ random network. In particular, lower densities of amorphous carbon would favor the network disconnection and, in consequence, lead to vanishing elastic constants [95]. The higher conductivity results are not surprising since the Tersoff potential gives a larger density than DFT and experiments for a given $sp^{3}$ content [50,96]. Recently, another set of simulations using Tersoff was reported [48].

Extending the interaction cut-off of the Tersoff potential allows for better agreement with DFT results regarding density versus $sp^{3}$ content and led to larger conductivity values above 2.8 g/cm${}^{3}$. It is interesting that, in our aC simulations, the thermal conductivity of NT reaches values higher than that of nanowires, and that of nanowires reaches values higher than that of bulk as clearly shown in Figure 11. The ratio of NT thermal conductivity with respect to NW and bulk values was found to vary between 1.1 and 1.5.

Figure 12 displays thermal conductivity versus density at 300 K, including the fit by Balandin et al. [31] for amorphous carbon nanofilms experiments. Our simulation results can be described fairly well by an upward shift of this fit. The fit to the bulk MD results from Suarez-Martinez and Marks [44] is also shown, giving a slightly lower slope; however, with a reasonable agreement with our results, as expected, since the same empirical potential and similar sample generation were used.

The MD results from Fujii [97] and Zhang et al. [42] using AIREBO and Tersoff potentials, respectively, show some agreement with our results, with Tersoff leading to a higher slope in conductivity versus density values. The results from Giri et al. [48], using the Tersoff potential with an extended cut-off, describe the high conductivity in the experiment from Morath et al. [34] but overestimate all other experimental values (Falabella et al. [98], Bullen et al. [30], Chen et al. [33] and Balandin et al. [31]), which are close to or slightly lower than our results.

Beyond simple scaling arguments based on Einstein or Debye models, differences in thermal conductivity could also be associated with changes in vibrational frequencies due to the existence of a surface in the NW [94,99] and a second surface in the NT. An analysis of the density of vibrational states was performed as in Section 2.5. Figure 13 shows the results obtained for the NW of 2 nm radius and for the NT of 2 and 1 nm of external and internal radii, with both 20% $sp^{3}$ and 60% $sp^{3}$, at 300 K.

The results are also compared with a “partial” $G{\left(\omega \right)}_{z}$ calculated using only the contributions of the velocity in the z direction. In our simulations no significant difference was found between the contribution due only to the z direction or between the NT and the NW, which could explain the increase in conductivity in the NT with respect to the NW. Figure 13 also shows somewhat similar peaks and distribution for different hybridization. The simulated VDOS in Figure 13 shows a continuum region of vibrational states characteristic of amorphous materials [7,72,100], with a broad “boson peak” below 30 THz and another broad peak below 70 THz.

The boundary between propagons and diffusons at the Ioffe–Regel crossover was located at 60 THz in MD simulations of taC [40]. In our simulations, the mid-height of the boson peak, typically giving the expected location of this boundary, also occurs near this value [7,100]. Kumagai [101] calculated the VDOS for aC, with 33% $sp^{3}$ and a density of 2 g/cm${}^{3}$, using tight-binding and obtained a similar distribution. Our VDOS is also similar to the VDOS for 25% $sp^{3}$ aC simulated with the Tersoff potential [42].

In addition, there is some agreement between the peak positions in our VDOS and the position of the sharp main peaks found in MD simulations of graphene, at 16.5 THz (∼500 cm${}^{-1}$) and 45.2 THz [76]. The VDOS remains nearly unchanged with temperature in the range studied here, with the peaks becoming slightly lower and smoother as temperature increases. This is expected, since we are well below melting in this study, and there are no structural changes in our temperature range, unlike studies for high temperature [102]. Finally, our VDOS agrees roughly with the one for MD simulations of hot and disordered diamond nanoparticles [102].

The VDOS can be related to the infrared (IR) and Raman spectra by some matrix elements which modify peak intensity [103] and by considering the quantum vibrational mode population using the Bose–Einstein distribution [40]. Kumagai et al. [101] observed some structure in the simulated boson peak at low frequencies, which could be deconvoluted into two peaks, at 400 cm${}^{-1}$ and 700 cm${}^{-1}$; however, their origin was not discussed. The experimental Raman spectra for aC shows a similar structure, and the 400 cm${}^{-1}$ peak has been related to $sp^{2}$ content and the graphitic contributions at 500 cm${}^{-1}$ downshifted by strain [104].

The possible presence of those two peaks in the low-frequency region in our simulated VDOS is somewhat clear in Figure 13 when comparing the peaks of the 60%
$sp^{3}$ (lower $sp^{2}$) with the 20%
$sp^{3}$ (higher $sp^{2}$) content samples spectra; for the latter, a displacement can be observed towards 400 cm${}^{-1}$, while the former is closer to 700 cm${}^{-1}$; however, further studies are required to clarify and quantify this.

At higher frequencies, the experimental Raman spectra for aC shows a graphitic (G) and a disordered (D) peaks, which shift and change intensity according to $sp^{3}$ content [51,105]. For 20% $sp^{3}$, the G-peak is observed near 1510 cm${}^{-1}$ and the D peak near 1300 cm${}^{-1}$ [51,105]. In our MD simulations, the broad high-frequency peak can also be deconvoluted into G and D peaks, with the G peak located as in experiments but with the D peak somewhat lower in frequency.

A good agreement was found with the experimental results by Lopinski et al. [106], for high $sp^{3}$, also showing three peaks. We note that peaks are also observed for low $sp^{3}$ content, in agreement with DTF results [107] but in contrast with the VDOS for the Tersoff potential with an extended cut-off [48], where no peaks are observed for 12.5% $sp^{3}$ (2.1 g/cm${}^{3}$). VDOS remains nearly unchanged with temperature in the range studied here, with the peaks becoming slightly lower and smoother as the temperature increases.

In the experiments by Zhan et al. [10], NT arrays with NT 85 nm diameter and 35 nm inner diameter had the lowest thermal conductivity, 0.023 Wm${}^{-1}$K${}^{-1}$. This is 100 times lower than our simulated results. The corresponding bulk density for the aC in the array would be ∼1.6 g/cm${}^{3}$, which effectively corresponds to $sp^{3}$ content of around 10% [51]; however, the array has a global porosity of around 94% for the above case. In order to explain the thermal conductivity of the array, they assume a bulk aC thermal conductivity of 0.150 Wm${}^{-1}$K${}^{-1}$, appropriate for such low $sp^{3}$ content [29], although there is significant variation in the experimental values.

In addition, they used Effective Medium Theory (EMT) to consider the joint conductivity of aC and air for a given global porosity and obtain excellent agreement with the experimental value. This approach neglects the thermal conductivity reduction due to phonon scattering at NT surfaces and also scattering at the junctions in the NT array.

Our simulated nanotubes are significantly smaller, have larger bulk density and a “porosity” of approximately 2%, 12% and 30%, compared to a NW. Therefore, the application of EMT would give a decrease in conductivity with respect to a NW. However, an array of our NT would be expected to display significantly lower conductivity than a single NT.

## 4. Summary and Conclusions

The thermal conductivity of amorphous carbon (aC) nanostructures is of technological interest; however, there are only a few simulation studies [40,41,42] and only a couple for a wide range of hybridization values [44,48]. All those studies are for bulk systems. Here, we presented studies for bulk and two types of nanostructures—nanowires (NW) and nanotubes (NT)—finding an unexpected increase in conductivity for those nanostructures, which is explained by the increased elastic modulus of the nanostructures.

In addition, the differences between NW and NT conductivity was explained in terms of changes in the elastic modulus with the surface-to-volume ratio as considered for the mechanical properties of metallic nanofoams [88]. This leads to an increased conductivity for NT, despite their porosity. All these results are expanded below. The thermal conductivity of aC nanostructures was obtained by non-equilibrium MD simulations with a temperature gradient. Simulated aC bulk samples satisfied a relationship between the density and hybridization degree and Young’s modulus that was compatible with the experimental data [32,51]. The simulated bulk thermal conductivity was somewhat larger than in experiments for aC nanofilms [31] or thicker films [30] but within the values from other experiments [33].

Nanowires (NW) and nanotubes (NT) were cut from bulk samples with different hybridization and relaxed to obtain samples without pre-stress. In this work, we focused on NW and NT with a 2 nm external radius and internal radii of 0.5, 1 and 1.3 nm for NT. The elastic modulus versus hybridization for the nanowires agrees well with experiments [31]. The thermal conductivity increased roughly linearly with the density and $sp^{3}$ content, similar to the bulk results but reaching higher values.

Neglecting quantum corrections, the thermal conductivity was found to be practically constant with temperature in the studied range for all cases. We considered a simple qualitative way to include quantum corrections [62,64], which suggests that the quantum corrections are small at 300 K or higher temperature and that a comparison between our classical simulations and room-temperature experiments would be reasonable. The bulk experimental results showed that conductivity increased with temperature up to 500 K [30].

However, our results are consistent with those of Balandin et al. [31] who found virtually constant conductivity for thin aC films above 150 K. Experiments for Si NW also found that the thermal conductivity was nearly constant with temperature as attributed to surface scattering. Donadio and Galli [108] found that NW of Si coated with an amorphous layer also had an almost constant behavior of conductivity with temperature.

A curious result was that the thermal conductivity of NT reached values higher than that of NW, and that of NW reached values higher than that of bulk. This is not what happens in crystalline solids [92], and it is not expected in amorphous solids. Malhotra and Maldovan discussed possible strategies to reduce thermal conductivity and include NT structures as an effective pathway to reach low conductivity [109].

Furthermore, in our study, an increase of thermal conductivity with the internal radius in NT was observed, in disagreement with calculations including surface scattering in NT [54]. However, we found that a small modification to the Einstein model was sufficient to predict our results based on the increases observed in the elastic moduli, which in turn, can be explained by a model used in nanofoams [88,89] based on the ratio between the surface area and the solid volume (S/V).

Future work might include the thermal conductivity dependence of the nanowires radii and temperatures much higher than those presented in this work, reaching above 1000 K. Bulk aC is stable up to extremely high temperatures, and perhaps a decrease in conductivity could be observed in simulations; however, it would be difficult to measure experimentally. It is known from experiments that aC films undergo a decrease of the elastic constants for temperatures over approximately 800 K, induced by internal stress relief [110]. This could lead to discontinuities or abrupt reductions of the thermal conductivity near and above that temperature.

Therefore, future simulations of aC nanostructures at higher temperatures could offer insights into their technological window of application for low thermal conductivity requirements. The role of strain and stress in the thermal properties and the vibrational density of states on these structures can also be explored further. Future studies might analyze in-depth the role of the heat transferred between the thermostat and the different nanostructures with varying cross-sections and include thermal conductance and thermal resistance in addition to the heat conductivity.

In order to gain insights into technological applications [10,11], larger NW/NT diameters would have to be considered alongside lower $sp^{3}$ values. The thermal conductivity of isolated NW/NT is required as well as the inclusion of junctions that might influence the thermal conductivity in NW foams and aerogels. 

## Figures and Tables

**Figure 1 nanomaterials-12-02835-f001:**
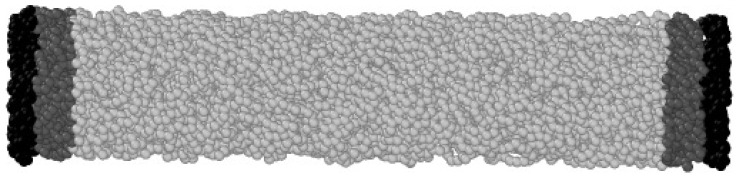
Snapshot of the system configuration used to calculate thermal conductivity. Colors represent different regions: fixed atoms (black), thermostat region (dark gray) and heat-flow region (light gray).

**Figure 2 nanomaterials-12-02835-f002:**
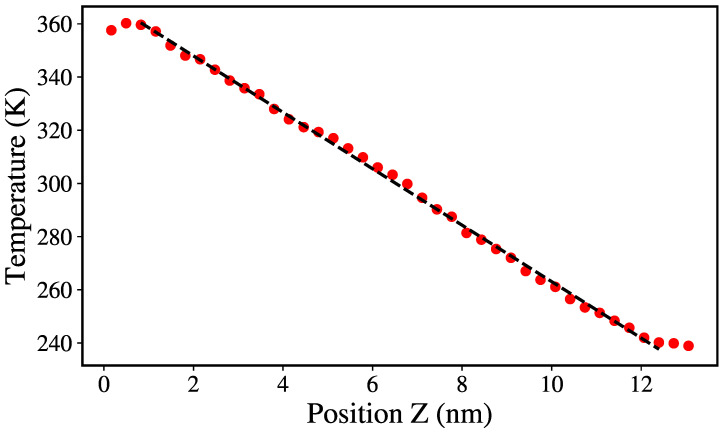
Temperature profile for a 20% $sp^{3}$ NT, with 1 and 2 nm of internal and external radii, at 300 K. The dotted line shows the linear fit of the temperature gradient.

**Figure 3 nanomaterials-12-02835-f003:**
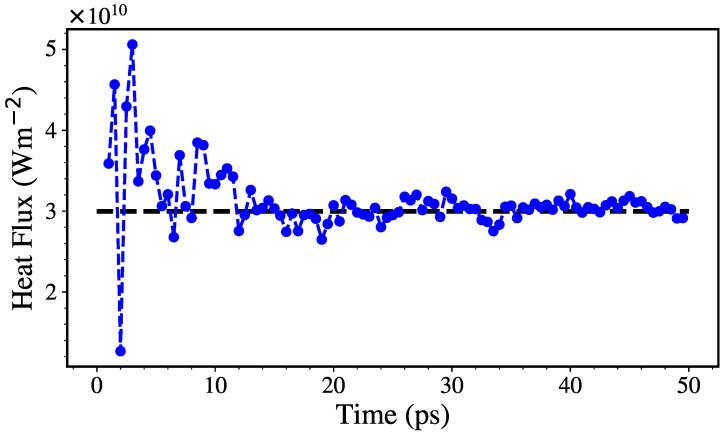
Heat flux vs. time for a 20% $sp^{3}$ NT with 1 and 2 nm of internal and external radii, at 300 K. The dashed line show the flux value use in the calculations.

**Figure 4 nanomaterials-12-02835-f004:**
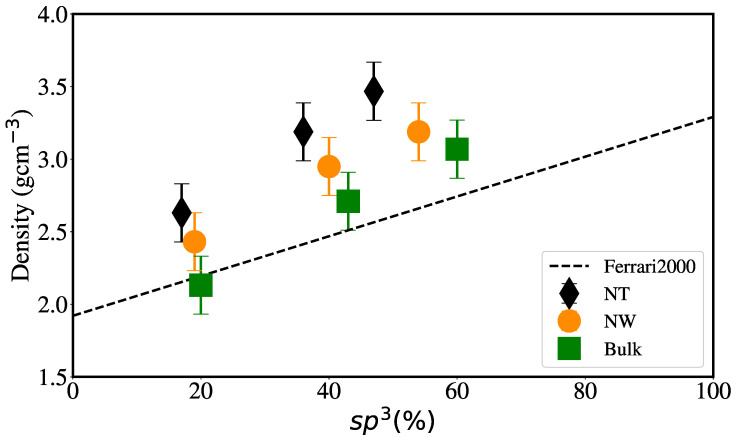
Sample density vs. $sp^{3}$ content. The dashed line shows the linear relationship from experiments by Ferrari et al. [32] for aC films.

**Figure 5 nanomaterials-12-02835-f005:**
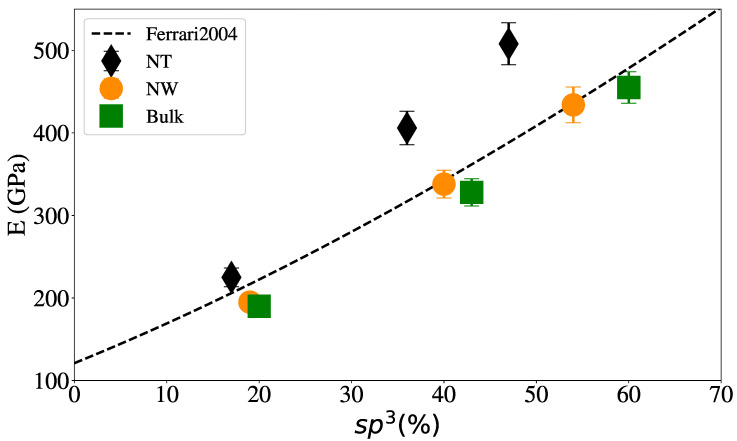
Elastic modulus vs. $sp^{3}$ content for NW with 2 nm of radius and NT with 2 nm of external radius and 1 nm of internal radius. Dashed line indicates the fit for aC thin-film experiments [86].

**Figure 6 nanomaterials-12-02835-f006:**
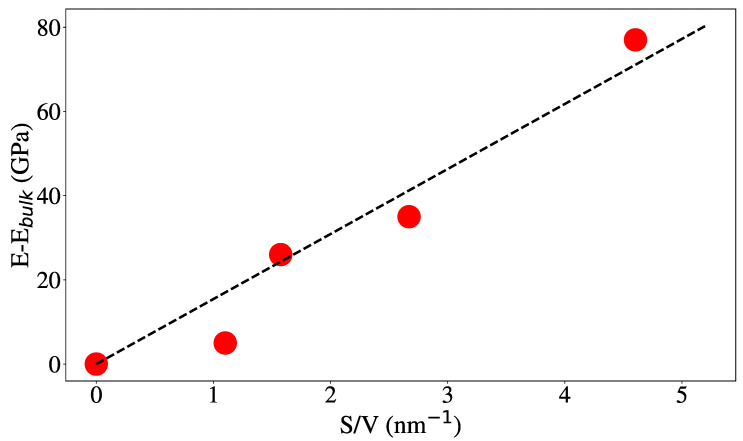
E vs. S/V for 20 $sp^{3}$ content. The results form the bulk, NW and NT of external radius of 2 nm, and internal radii of 0.5, 1.0 and 1.3 nm are shown from left to right.

**Figure 7 nanomaterials-12-02835-f007:**
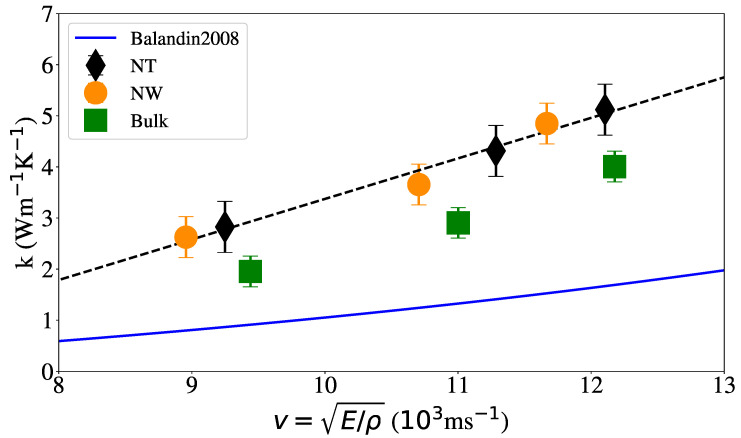
Thermal conductivity at 300 K vs. sound velocity $v=\sqrt{E/\rho }$ for different geometries (NT internal radius is 1 nm). The solid line shows the fit obtained from aC thin-film experiments by Balandin et al. [31], where k=−2.82+1.77ρ and $k=0.04E^{2/3}-0.4$ were used. The dashed line shows a linear fit to our MD results, k=−4.55+0.79v.

**Figure 8 nanomaterials-12-02835-f008:**
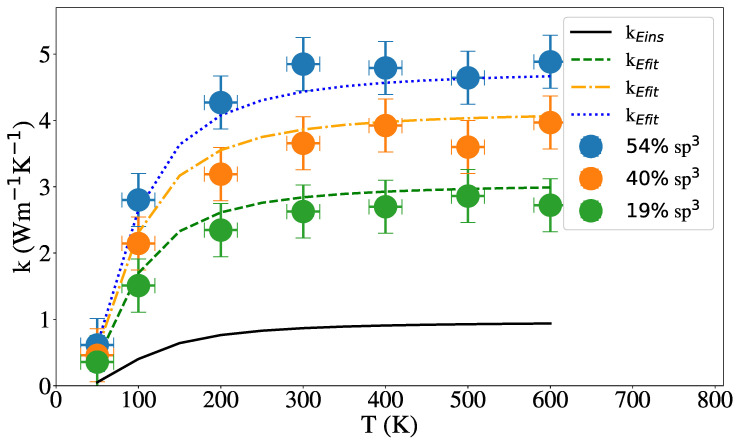
Thermal conductivity vs. temperature for NW with three different hybridization values. The solid line represent the Einstein limit given by Equation (Equation 3), the dashed, dash-dotted and dotted lines represent Equation (Equation 6) for the NW of 19%, 40% and 54% of $sp^{3}$ hybridization respectably.

**Figure 9 nanomaterials-12-02835-f009:**
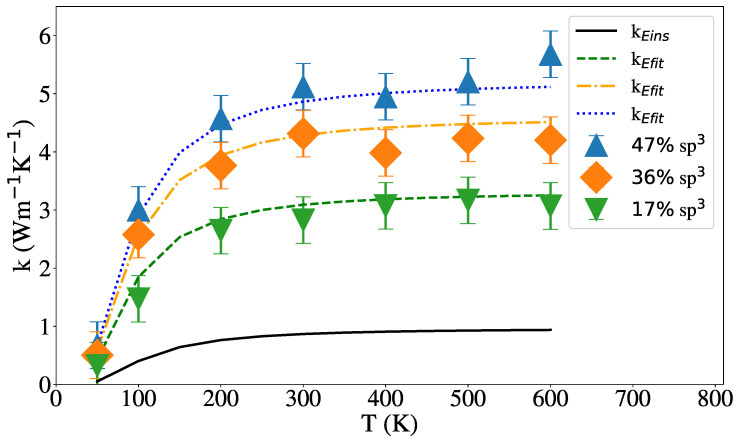
Thermal conductivity vs. temperature for NT with 2 and 1 nm of external and internal radius, for different percent of $sp^{3}$ hybridization. The solid line represent the Einstein limit given by Equation (Equation 3), the dashed, dash-dotted and dotted lines represent Equation (Equation 6) for the NT of 17%, 36% and 47% of $sp^{3}$ hybridization, respectively.

**Figure 10 nanomaterials-12-02835-f010:**
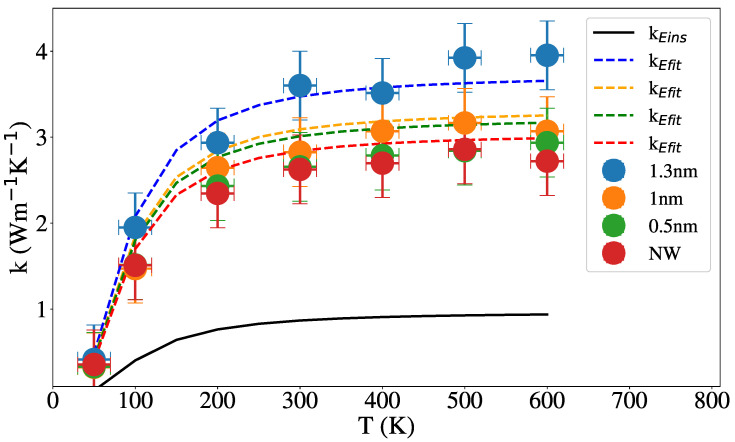
Thermal conductivity vs. temperature for NT with 20% $sp^{3}$ and 2 nm external radius for different internal radii. NW results are also included.

**Figure 11 nanomaterials-12-02835-f011:**
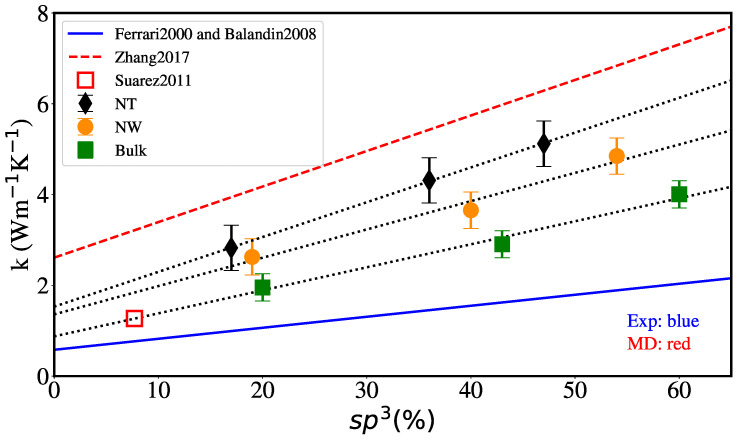
Thermal conductivity at 300 K vs. the percent of $sp^{3}$ hybridization for different geometries (the NT internal radius is 1 nm). Dotted lines show linear fits for the current MD values. There is also a MD result from [44], and the dashed line shows the fit to MD results using the Tersoff potential [42]. The solid line shows the linear relationship obtained from fits to experiments for aC thin films in Refs. [31,32].

**Figure 12 nanomaterials-12-02835-f012:**
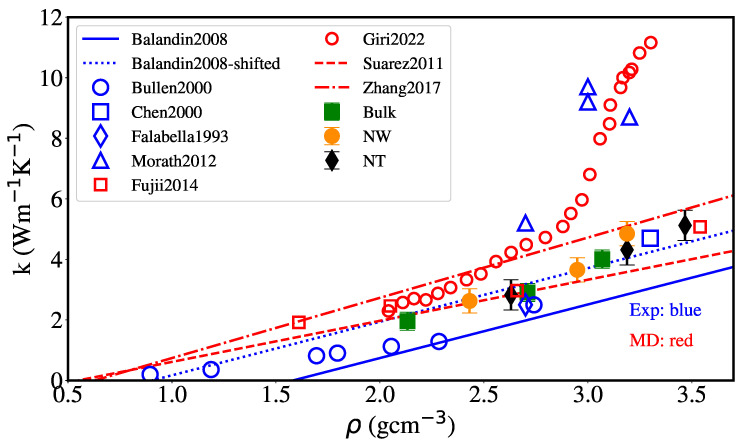
Thermal conductivity at 300 K vs. density ρ for different geometries (NT internal radius is 1 nm). The solid line shows the fit to aC thin-film experiments by Balandin et al. [31], k=−2.82+1.77ρ. The dashed line shows the fit to the MD results by Suarez-Martinez and Marks [44], k=−0.74+1.36ρ, and the dotted-dashed black line shows the fit to MD results using the Tersoff potential [42], k=−1.25+1.99ρ. The dotted line shows the previous fit by Balandin et al. shifted 0.8 Wm${}^{-1}$K${}^{-1}$ upwards. Experimental results in blue include circles [30], empty squares [33], diamonds [98] and triangles [34], and are shown together with MD results in red, including empty squares for AIREBO [97] and circles for extended Tersoff [48].

**Figure 13 nanomaterials-12-02835-f013:**
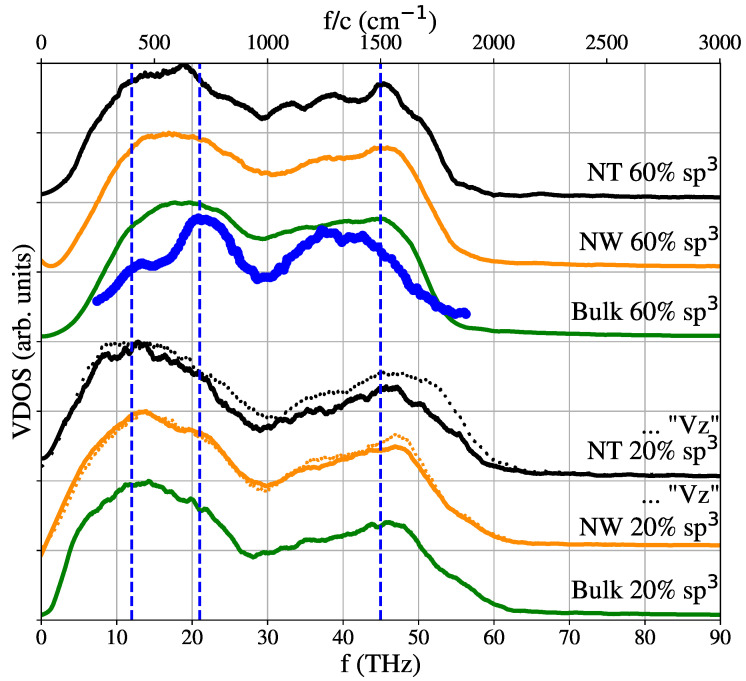
Vibrational density of states for bulk, NW and NT. 20% $sp^{3}$ shows no significant preferences in any particular direction, as shown by considering only *z* velocities, along the NW/NT axis. For 60% $sp^{3}$ hybridization, there are no significant changes that could explain the increase in their thermal conductivity. Dashed vertical blue lines representing 400 cm${}^{-1}$, 700 cm${}^{-1}$ and 45.2 THz were placed to help guide the eye. The blue solid line represents the experimental values obtained by Lopinski et al. [106].

## Data Availability

The data that supports the findings of this study are available from the corresponding author upon reasonable request.

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
