# Peer review of "Enhancing the Thermal Conductivity of Amorphous Carbon with Nanowires and Nanotubes"

_nanomaterials, 2022, doi:10.3390/nano12162835_

Round 1
Reviewer 1 Report
The manuscript evaluates the thermal conductivity of amorphous carbon nanowires with a 2 nm radius, and amorphous carbon nanotubes with 0.5, 1 and 1.3 nm internal radius and 2 nm external radius using atomistic classical Molecular Dynamics simulations.
Simulations are performed with the software LAMMPS.
The topic falls in the journal scope. The manuscript is organized with figures. However, the reviewer has some reservations and believes a revision is needed to address the following concerns before it can be accepted for publication.
1. Novelties of the paper should be better pointed out since it is not clear the new contributions of the manuscript.
Reviewer 2 Report
The authors report a computational study, based on Atomistic MD simulations, on the thermal conductivity of amorphous CNTs and NW.The concept is good and the overall computational protocol is consistent.
My feeling is that the reported data match with the broad interest of the journal and are worthy of publication.
A few issues are given as follows:
1. Some comparison should be given between amorphous and non-amorphous (data could be retrieved from literature or similar studies).
2. The elevation of T is provided in a constant rate (is the gradient of T constant )?
3. How it is ensured that by T increase no substantial structural deformations are not taking place (at least it was not obvious to me) ? This could probably distort (in a minor/major way the bonding characteristics)
4. The discussion of the T effect on the phonon spectra is somehow dense leading to some misunderstandings. For example, the physical origin of why the VDOS remains unaltered with T is not fully understood.
5. In the conclusions section one stress the differences due to the size of the aCNTS and T.
Round 2
Reviewer 1 Report
The manuscript seems a collection of known results. It is then performed a numerical analysis with the existing software LAMMPS.
Reviewer 2 Report
The revised version has been improved and can be published in its present form.